# Four-Dimensional Variational Data Assimilation and Sensitivity of Ocean Model State Variables to Observation Errors

Victor Shutyaev [1,*], Vladimir Zalesny [1], Valeriy Agoshkov [1,2], Eugene Parmuzin [1,2] and Natalia Zakharova [1]

[1] Marchuk Institute of Numerical Mathematics, Russian Academy of Sciences, 119333 Moscow, Russia; vzalesny@yandex.ru (V.Z.); agoshkov@inm.ras.ru (V.A.); e.parmuzin@inm.ras.ru (E.P.); zakharova_nb@inm.ras.ru (N.Z.)
[2] Faculty of Computational Mathematics and Cybernetics, Lomonosov Moscow State University, 119991 Moscow, Russia
* Correspondence: victor.shutyaev@mail.ru; Tel.: +7-495-984-8128

**Abstract:** The use of Four-Dimensional variational (4D-Var) data assimilation technology in the context of sea dynamics problems, with a sensitivity analysis of model results to observation errors, is presented. The technology is applied to a numerical model of ocean circulation developed at the Marchuk Institute of Numerical Mathematics, Russian Academy of Sciences (INM RAS), with the use of the splitting method and complemented by 4D-Var data assimilation with covariance matrices of background and observation errors. The variational data assimilation involves iterative procedures to solve inverse problems so as to correct sea surface heat fluxes for the model under consideration. An algorithm is formulated to study the sensitivity of the model outputs, considered as output functions after assimilation, to the observation errors. The algorithm reveals the regions where the output function gradient is the largest for the average sea surface temperature (SST) in a selected area, obtained by assimilation. In the numerical experiments, a 4D variational problem of SST assimilation for the Baltic Sea area is solved.

**Keywords:** variational data assimilation; ocean circulation model; sea surface temperature; observations; sensitivity analysis

## 1. Introduction

In recent years, there has been growing interest in observational data assimilation problems for geophysical hydrodynamics models, due to advances in the creation of more and more powerful computing systems and the development of new measuring technologies, as well as new methods and numerical algorithms. The combination of observational data and hydrodynamic forecasts is very important for computational technologies that model and analyze natural phenomena. The methods of data assimilation (DA) link model calculations to real data in order to construct, or refine, unknown inputs and/or parameters and to improve the accuracy of forecasts [1–11].

Computational technologies for forecasting and monitoring involve the use of numerical models of ocean dynamics to simulate hydrodynamic processes in order to investigate the structure and variability of hydrophysical fields. Ocean circulation mathematical models are set up on complex nonlinear partial differential equations which describe the behavior of hydrophysical fields of pressure, water density, sea surface level, temperature, salinity and currents. To solve problems in these fields requires the development of efficient numerical methods [12–16].

To assimilate observational data in geophysical hydrodynamic models, two main approaches are currently widely used: statistical and variational. The statistical approach uses the methods of the theory of probability and is related to the least square method,

introduced historically by Gauss (1795) and Legendre (1805), and rigorously justified by Markov [17] and Kolmogorov [18]. This approach inspired the optimal interpolation methods and Kalman filter methods used in many applications [1–4]. It should be noted that Optimal Interpolation is equivalent to a specific variational assimilation problem, as demonstrated in [19].

The variational approach uses the calculus of variations and optimal control [5,20], involving the adjoint equations theory [4,6–8,11,16]. The main idea of this approach is to minimize a cost function related to the observational data, in the subspace of model solutions. This facilitates solving initialization problems, so as to study the sensitivity of the response functions, and to estimate model parameters. For time-dependent problems, four-dimensional variational assimilation (4D-Var) is usually applied [1–11]. Particular attention is paid to 4D-Var ocean modeling [21–25].

In this article, we continue the research reported in [25] on 4D-Var data assimilation technology in solving ocean dynamics problems. This technology is applied to the numerical ocean circulation model, INMOM, developed at the Marchuk Institute of Numerical Mathematics, Russian Academy of Sciences, and described by primitive equations under hydrostastics and Boussinesq approximations in the sigma-coordinate system. The main feature of the INM RAS model, in comparison with other ocean models (e.g., [12–14]), is that its numerical realization uses the splitting method, taking into account different physical processes and directions of spatial coordinates [26,27]. The splitting method helps to split the hydrodynamics model into simpler problems, which are subsequently solved, in time, with the use of explicit or implicit schemes [16,27,28]. For variational assimilation of observational data, the problem of cost function minimization is reduced to a coupled system of original and adjoint equations in a four-dimensional time–space domain, to be solved in forward and backward time, respectively. This system is called an optimality system, and is solved by efficient iterative algorithms with properly chosen iterative parameters [25]. The novelty of this paper is the use of background and observation error covariance matrices in the cost function to be minimized, and the study of the sensitivity of model results to observation errors. One of the achievements of this study is the provision of an efficient method of heat flux correction, which was a significant problem in the early coupled ocean–atmospheric model, as seen in, for example, [29]. This is an important result which could help in the development of climate and Earth system models.

In the process of variational data assimilation, the solutions of the optimality system depend on the observational data, which often contain errors. An important issue is the study of the sensitivity of the model results, obtained after assimilation, to the errors of observational data. Model outputs are of interest in the form of the model variables (temperature, salinity, etc.) and their functions or functionals. It is worth estimating the sensitivity of the model results to the observation errors when the model output is obtained from the optimality system after variational assimilation. The optimality system is constructed with the use of the necessary optimality condition, which means that the cost function gradient equals zero. To investigate the model output sensitivity, the differentiation of the optimality system is required, with respect to observational data, which, therefore, gives rise to the so-called second-order adjoint problem [30]. Use of the second-order adjoint to study the sensitivity of an output function to model parameters in the process of initialization was first carried out in [31]. The forecast sensitivity, with respect to observational data, was considered, in [32], for a 4D-Var data assimilation problem. Using the results of [32], methods for estimating observational impact on 4D-Var were studied in [33] application to shallow-water equations. Sensitivity of an ocean model output to the uncertainties in the input data was investigated by modern 3D DA methods in a number of studies [3,34–38]. The optimal solution sensitivity for 4D-Var data assimilation problems is related to statistical properties [39–42]. For the initialization problem, the output function sensitivity, with respect to observational data, was studied in [43], in the general form for a nonlinear dynamic model. In the present article, an algorithm is developed to estimate the sensitivity of the model results, considered as output functions after variational assimilation,

with respect to observation errors. The algorithm may help to reveal the regions where the gradient of the output function is the largest, for the averaged sea surface temperature in a selected area, obtained by assimilation. It is important to know the regions with a large gradient in the output function because the model output is most sensitive to observation errors in these regions, which is useful in understanding the propagation of errors in ocean predictions.

The article is organized as follows. Section 2 presents the mathematical model of ocean circulation, described by primitive equations, and the steps of the splitting method for its approximation. In Section 3, the 4D-Var data assimilation technique is presented with the use of background and observation error covariances to correct sea surface heat fluxes. In Section 4, the output function sensitivity to observation errors is considered, and an algorithm is formulated to construct the output function gradient, with the use of the cost function Hessian. Section 5 includes the results and discussion of numerical experiments for the Baltic Sea water area with the use of the presented technology. The conclusion contains the main results.

## 2. Mathematical Model of Ocean Circulation and Numerical Methods

The system of primitive equations of ocean hydrodynamics in geographical coordinates is considered in the domain $D \in \mathbf{R}^3$, under hydrostatics and Boussinesq approximations [44,45]:

$$
\begin{cases}
\dfrac{d\vec{u}}{dt} + \begin{bmatrix} 0 & -f \\ f & 0 \end{bmatrix} \vec{u} - g\,\mathbf{grad}\zeta + A_u\vec{u} + (A_k)^2\vec{u} = \vec{F} - \dfrac{1}{\rho_0}\mathbf{grad}P_a - \dfrac{g}{\rho_0}\mathbf{grad}\displaystyle\int_0^z \rho_1(T,S)dz', \\[4mm]
\dfrac{dS}{dt} + (\bar{U},\mathbf{Grad})S = -A_S S + f_S, \quad \dfrac{dT}{dt} + (\bar{U},\mathbf{Grad})T = -A_T T + f_T, \\[4mm]
\dfrac{\partial\zeta}{\partial t} - m\dfrac{\partial}{\partial x}\Big(\displaystyle\int_0^H \Theta(z)u\,dz\Big) - m\dfrac{\partial}{\partial y}\Big(\displaystyle\int_0^H \Theta(z)\dfrac{n}{m}v\,dz\Big) = f_3,
\end{cases}
\tag{1}
$$

where $(x,y,z) \in D$, $t \in (0,\bar{t})$, $\bar{U} = (u,v,w)$ is the velocity vector, $\vec{u}=(u,v)$, $S$ is the salinity, $T$ is the temperature, $\zeta$ is the sea surface level, $\rho_1(T,S) = \rho_0\beta_T(T - T^{(0)}) + \rho_0\beta_S(S - S^{(0)}) + \gamma\rho_0\beta_{TS}(T,S)$ is the water density, $\vec{F} = (F_1, F_2)$ is the forcing, $P_a$ is the atmospheric pressure, $f_T$, $f_S$ are the functions of the 'internal' sources, $\rho_0 = const \approx 1$ is the mean density, $\beta_{TS}(T,S)$ is the sum of all other terms of the expansion of the function of state $\rho = \rho(T,S)$, $T^{(0)}$, $S^{(0)}$ are given reference values, $f_3 \equiv f_3(x,y,t)$ is the function related to the tide-generating forces, $A_\varphi\varphi \equiv -\mathbf{Div}(\hat{a}_\varphi\mathbf{Grad}\varphi)$, $\beta_T, \beta_S, \gamma, g = const$, $n = 1/r$, $m = 1/(r\cos y)$, $r = R - z \approx R$, $\Theta(z) \equiv (R - z)/R \approx 1$, $R$ is the Earth radius.

The operators $A_{\varphi}\varphi \equiv -\mathbf{Div}(\hat{a}_\varphi\mathbf{Grad}\varphi)$ involve $\hat{a}_\varphi = diag((a_\varphi)_{ii})$, where $(a_\varphi)_{11} = (a_\varphi)_{22} \equiv \mu_\varphi$, $(a_\varphi)_{33} \equiv \nu_\varphi$, and $\varphi$ may take the values $u, v, T, S$. We assume that $\mu_u = \mu_v \equiv \mu$, $\nu_u = \nu_v \equiv \nu$, and $\mu, \nu, \mu_T, \mu_S, \nu_T, \nu_S$ are diffusion coefficients that are supposed to be positive bounded functions. The fourth order operator $(A_k)^2$, with $A_k$ taken for $A_\varphi = A_k$, is defined by the matrix $\hat{k} = diag\{k_{ii}\}$ with non-negative diagonal elements $k_{ii}$ that are the viscosity coefficients in the respective directions. We consider $f = f(u) = l + mu\sin y \equiv l + f_1(u)$, where $l = l(y)$ is the Coriolis parameter $l = 2\omega\sin y$, and $\omega$ is the Earth angular rotaton speed.

The system (1) is considered in $D \times (0,\bar{t})$ with the corresponding boundary and initial conditions [25,45]. For example, the boundary conditions on the sea surface $\Gamma_S = \Omega$ for $T$ and $S$ are the following:

$$
\begin{cases}
-\nu_T\dfrac{\partial T}{\partial z} = \gamma_T(T_a - T) + Q_T, \\[4mm]
-\nu_S\dfrac{\partial S}{\partial z} = \gamma_S(S_a - S) + Q_S,
\end{cases}
\tag{2}
$$

where $\nu_T, \nu_S$ are the turbulent exchange coefficients, $\gamma_T, \gamma_S$ are the relaxation coefficients, $Q_S$ and $Q_T$ are the surface salinity and heat fluxes, $Q_T$ represents the sum of shortwave

and longwave radiations and latent heat fluxes, $T_a$ and $S_a$ are the specified values of $T$ and $S$. More details on boundary conditions are given in [25].

The vertical velocity $w$ is related to $u$ and $v$ by [25]:

$$w(x,y,z,t) = \frac{1}{r}\left( m\frac{\partial}{\partial x}\left( \int_z^H rudz' \right) + m\frac{\partial}{\partial y}\left( \frac{n}{m} \int_z^H rvdz' \right) \right), \quad (x,y,t) \in \Omega \times (0,\bar{t}). \quad (3)$$

Initial conditions are defined by

$$u = u^0, v = v^0, S = S^0, T = T^0, \zeta = \zeta^0, \text{ for } t = 0, \quad (4)$$

where $u^0$, $v^0$, $S^0$, $T^0$, $\zeta^0$ are given functions.

The system (1)–(4) to find the functions $u, v, S, T, \zeta$, and then $w$ by (3), is considered below to describe large-scale ocean dynamics.

The numerical INM RAS ocean dynamics model for (1)–(4) uses the splitting method [26,27] and the $\sigma$-coordinate system [27,28]. The importance of these two components for 4D-Var was discussed in [25].

The splitting method is used for the approximation of the problem (1)–(4) in time, facilitating a split of the hydrodynamics model into simpler problems that are subsequently solved, in time, with the use of explicit or implicit schemes. Consider the time grid on the interval $[0;\bar{t}]$: $0 = t_0 < t_1 < \ldots < t_{J-1} < t_J = \bar{t}$, with the time steps $\Delta t_j = t_j - t_{j-1}$, and problem (1)–(4) on $(t_{j-1}, t_j)$, under the condition that the solution $u_k, v_k, \xi_k, T_k, S_k$ at the intervals $(t_{k-1}, t_k)$, $k = 1, 2, \ldots, j-1$, is already determined. Then, the splitting method is applied in the form of the following steps.

**Step 1.** Solve the equation for the temperature:

$$T_t + (\bar{U}, \mathbf{Grad})T = \mathbf{Div}(\hat{a}_T \cdot \mathbf{Grad}\ T) + f_T \text{ for } D \times (t_{j-1}, t_j). \quad (5)$$

**Step 2.** Solve the equation for the salinity:

$$S_t + (\bar{U}, \mathbf{Grad})S = \mathbf{Div}(\hat{a}_S \cdot \mathbf{Grad}\ S) + f_S \text{ for } D \times (t_{j-1}, t_j). \quad (6)$$

**Step 3.** Solve the system:

$$\begin{cases} \underline{u}_t + \begin{bmatrix} 0 & -f \\ f & 0 \end{bmatrix}\underline{u} - g\,\mathbf{grad}\xi + A_u\underline{u} + (A_k)^2\underline{u} = \\ \\ = \vec{F} - \frac{1}{\rho_0}\mathbf{grad}\left( P_a + g\int_0^z \rho_1(\bar{T},\bar{S})dz' \right) \text{in } D \times (t_{j-1}, t_j), \\ \\ \xi_t - \mathbf{div}\left( \int_0^H \Theta\underline{u}dz \right) = f_3 \text{ in } \Omega \times (t_{j-1}, t_j), \\ \underline{u} = \underline{u}_{j-1}, \ \xi = \xi_{j-1} \text{ for } t = t_{j-1}, \ \underline{u}_j \equiv \underline{u}(t_j) \text{ in } D \end{cases} \quad (7)$$

and take the functions $\underline{u}$ and $\xi_j$ as approximations to the exact $\vec{u}$ and $\xi$ on $(t_{j-1}, t_j)$.

Problems (5)–(7) may also be split by taking into account the directions of spatial coordinates [27,46]. Other numerical methods and details for solving the sub-problems (5)–(7) may be found in [44,46].

More detailed information on the numerical INMOM model is given in [46–48].

## 3. Four-Dimensional Variational SST Assimilation

The 4D-Var data assimilation involves a minimization of the cost function related to observations on the model solutions, with the aim to update the state variables

Hereinafter, the heat flux $Q = -\nu_T \dfrac{\partial T}{\partial z}$ on $\Omega$ is supposed to be unknown and taken as a control function to minimize the cost function

$$J(Q) = \frac{1}{2}\int\limits_{0}^{\bar{t}}\int\limits_{\Omega}(Q - Q^{(0)})\mathcal{B}^{-1}(Q - Q^{(0)})d\Omega dt + \frac{1}{2}\sum_{j=1}^{J}J_{0,j},$$

$$J_{0,j} \equiv \int\limits_{t_{j-1}}^{t_j}\int\limits_{\Omega}(T|_{z=0} - T_{obs})\mathcal{R}^{-1}(T|_{z=0} - T_{obs})d\Omega dt,$$

(8)

where $T_{obs}$ is the SST observation function given on $\Omega$, $Q^{(0)}$ is an initial approximation (a background) for $Q$, the operators $\mathcal{B}$ and $\mathcal{R}$ are the background and the observation error covariance operators. In our experiments, for $Q^{(0)}$ the mean climatic flux is used (see Section 5).

Finding the heat flux $Q$ and the solution to problem (1)–(4) that gives the minimum cost function (8) is the 4D variational data assimilation problem. To solve this 4D-Var assimilation problem, we use the necessary optimality condition and reduce the problem to the optimality system, which involves the original problem (1)–(4) and the adjoint equation, with the condition

$$T^* + \mathcal{B}^{-1}\left(Q - Q^{(0)}\right) = 0 \ \text{ on } \ \Omega,$$

(9)

obtained from the fact that the gradient of the cost function, with respect to the heat flux $Q$, is equal to zero. The solution $T^*$ of the adjoint problem at Step 1 of the splitting method is defined by

$$T^*_t + \mathbf{Div}(\bar{U}T^*) + \mathbf{Div}(\hat{a}_T \cdot \mathbf{Grad}\ T^*) = 0 \ \text{ for } \ D \times (t_{j-1}, t_j),$$

$$T^* = 0 \ \text{ if } \ t = t_j,$$

(10)

$$-\nu_T\frac{\partial T^*}{\partial z} = \mathcal{R}^{-1}(T|_{z=0} - T_{obs}), \ \ (x,y,t) \in \Omega \times (t_{j-1}, t_j).$$

(11)

Problems (10) and (11) are adjoint for (5), and should be solved backward in time. Note also that, in this case, the observation data $T_{obs}$ enter the right-hand side in the boundary condition (11).

To solve the above-formulated problem on correcting the heat flux $Q$ by the 4D-Var assimilation of SST data $T_{obs}$, iterative algorithms may be used. To formulate one of the algorithms, suppose that we have constructed an approximation $Q_{(k)}$ for $Q$ on $(t_{j-1}, t_j)$. One should then solve the original problem with $Q = Q_{(k)}$, find $T = T_{(k)}$, and solve the adjoint problem (10) and (11). Then, the next approximation $Q_{(k+1)}$ is obtained by:

$$Q_{(k+1)} = Q_{(k)} - \alpha_k(T^* + \mathcal{B}^{-1}(Q_{(k)} - Q^{(0)})), \ \ (x,y,t) \in \ \Omega \times (t_{j-1}, t_j),$$

(12)

where the parameters $\alpha_k$ are properly chosen, to ensure the convergence of the iterative algorithm [49].

Let us note that the background error covariance operator $\mathcal{B}$ enters the right-hand side of the iterative process (12), as well as the optimality condition (9).

## 4. Sensitivity of Model Outputs to Observation Errors

The solutions of the optimality system depend on the observational data, which can have errors. An important issue is the study of the sensitivity of the model results, obtained after assimilation, to the errors of observational data. We consider a model result in the form of a real-valued output function $G(T)$ of the sea temperature $T$. An example of such an output function is given below. An important issue is the sensitivity of the output function $G(T)$ to the observations $T_{obs}$, when $T$ is found as a result of assimilation from the

optimality system (5), (9)–(11). It is known from [7,50] that the sensitivity of the function $G(T)$ is given by the gradient of $G(T)$ with respect to $T_{obs}$:

$$\frac{dG}{dT_{obs}} = \frac{\partial G}{\partial T} \frac{\partial T}{\partial T_{obs}}. \tag{13}$$

Let $\delta T_{obs}$ be a variation of the function $T_{obs}$. Then, from (5), (9)–(11) we obtain the optimality system for the variations $\delta Q, \delta T, \delta T^*$:

$$\delta T_t + (\bar{U}, \mathbf{Grad})\delta T = \mathbf{Div}(\hat{a}_T \cdot \mathbf{Grad}\, \delta T) \quad \text{for } D \times (t_{j-1}, t_j),$$

$$\delta T = 0 \quad \text{if } t = t_{j-1}, \tag{14}$$

$$-\nu_T \frac{\partial \delta T}{\partial z} = \delta Q, \quad (x, y, t) \in \Omega \times (t_{j-1}, t_j),$$

$$-\delta T^*_t - \mathbf{Div}(\bar{U}\delta T^*) = \mathbf{Div}(\hat{a}_T \cdot \mathbf{Grad}\, \delta T^*) \quad \text{for } D \times (t_{j-1}, t_j),$$

$$\delta T^* = 0 \quad \text{if } t = t_j, \tag{15}$$

$$-\nu_T \frac{\partial \delta T^*}{\partial z} = \mathcal{R}^{-1}(\delta T|_{z=0} - \delta T_{obs}), \quad (x, y, t) \in \Omega \times (t_{j-1}, t_j),$$

$$\mathcal{B}^{-1}\delta Q + \delta T^* = 0 \quad \text{on } \Omega. \tag{16}$$

System (14)–(16) may be written as a variational data assimilation problem to determine $\delta T, \delta Q$ with observational data $\delta T_{obs}$. Excluding $\delta T, \delta T^*$ from the system (14)–(16), we show that it is equivalent to the equation for $\delta Q$:

$$\mathcal{H}\delta Q = \mathcal{C}\delta T_{obs}, \tag{17}$$

where $\mathcal{H}$ is the Hessian of the cost function $J(Q)$ from (8), and the operator $\mathcal{C}$ is defined, on the functions $\delta T_{obs}$, by the formula

$$\mathcal{C}\delta T_{obs} = \theta^*, \tag{18}$$

with $\theta^*$ being the solution of the adjoint problem

$$-\theta^*_t - \mathbf{Div}(\bar{U}\theta^*) = \mathbf{Div}(\hat{a}_T \cdot \mathbf{Grad}\, \theta^*) \quad \text{for } D \times (t_{j-1}, t_j),$$

$$\theta^* = 0 \quad \text{if } t = t_j, \tag{19}$$

$$-\nu_T \frac{\partial \theta^*}{\partial z} = \mathcal{R}^{-1}\delta T_{obs}, \quad (x, y, t) \in \Omega \times (t_{j-1}, t_j).$$

Note that the Hessian matrix $\mathcal{H}$ in (17) is a square matrix of second-order partial derivatives of the scalar field $J(Q)$, with the independent variables of the derivatives being latitude and longitude.

From (17),

$$\delta Q = \mathcal{H}^{-1}\mathcal{C}\delta T_{obs}. \tag{20}$$

Formula (20) relates the variation $\delta Q$ of the heat flux to the variations $\delta T_{obs}$ of the observational data. This formula may be used to estimate the sensitivity of the output function to observation errors.

The value of the gradient (13) on the variation $\delta T_{obs}$ is defined by

$$\left(\frac{dG}{dT_{obs}}, \delta T_{obs}\right) = \left(\frac{\partial G}{\partial T}, \delta T\right) = (\mathcal{F}, \delta Q), \tag{21}$$

where $\mathcal{F} = \phi^*|_{z=0}$, with $\phi^*$ is the solution of the adjoint equation

$$-\phi^*{}_t - \mathbf{Div}(\bar{U}\phi^*) = \mathbf{Div}(\hat{a}_T \cdot \mathbf{Grad}\,\phi^*) + \frac{\partial G}{\partial T} \quad \text{for} \ \ D \times (t_{j-1}, t_j),$$

$$\phi^* = 0 \quad \text{if} \ \ t = t_j,$$

(22)

and $(\cdot, \cdot)$ denotes a scalar product.

From (20) and (21), we obtain

$$\left( \frac{dG}{dT_{obs}}, \delta T_{obs} \right) = (\mathcal{F}, \mathcal{H}^{-1}\mathcal{C}\delta T_{obs}) = (\mathcal{C}^*\mathcal{H}^{-1}\mathcal{F}, \delta T_{obs}),$$

(23)

where $\mathcal{C}^*$ is the operator adjoint to $\mathcal{C}$. Therefore, the gradient, with respect to $T_{obs}$, is defined by

$$\frac{dG}{dT_{obs}} = \mathcal{C}^*\mathcal{H}^{-1}\mathcal{F}.$$

(24)

For the numerical implementation of the algorithm to calculate the output function gradient, according to formula (24), the following steps should be performed:

(1) solve the adjoint equation (22) and define $\mathcal{F} = \phi^*|_{z=0}$,
(2) solve the equation $\mathcal{H}\delta Q = \mathcal{F}$,
(3) solve the forward problem (14),
(4) compute the gradient by the formula:

$$\frac{dG}{dT_{obs}} = \mathcal{R}^{-1}\delta T|_{z=0}.$$

(25)

Having the derivative $\frac{\partial G}{\partial T}$, this algorithm may be used to estimate the sensitivity of the model output obtained as a result of the 4D-Var data assimilation.

## 5. Numerical Experiments for the Baltic Sea Area: Results and Discussion

The INM RAS numerical model of the Baltic Sea circulation [51] is used for numerical experiments. It is complemented by the procedures of variational data assimilation using the background and observation error covariance matrices. The boundary conditions, modified in accordance with [27], are considered.

The Baltic Sea water area was chosen for the calculations and to test the 4D-Var technology presented. The model computational area lay from 9.375° E to 30.375° E and from 53.625° N to 65.9375° N. The numerical spatial grid steps were 1/16 and 1/32 degrees in longitude and latitude, respectively, and 25 sigma-levels in depth. The time step was 5 min. Figure 1 presents the domain and the topography of the Baltic Sea. The results of previous model runs were used as initial conditions to start the assimilation experiment. The assimilation of observational data was carried out according to (5), (9)–(12) for a given assimilation window.

The coefficients of horizontal viscosity and diffusion of the second order in the calculations were taken to be equal to $2 \cdot 10^1$ m$^2$/s and $2 \cdot 10^2$ m$^2$/s, respectively. To set the vertical viscosity and diffusion, the Pacanowski–Philander parametrization was used [52]. The values of the coefficients of vertical viscosity and diffusion were limited by the minimum and maximum values. The vertical diffusion coefficient changed from a background value of $2 \cdot 10^{-6}$ m$^2$/s to $5 \cdot 10^{-3}$ m$^2$/s, and the vertical viscosity coefficient from $10^{-5}$ m$^2$/s, to $7 \cdot 10^{-3}$ m$^2$/s.

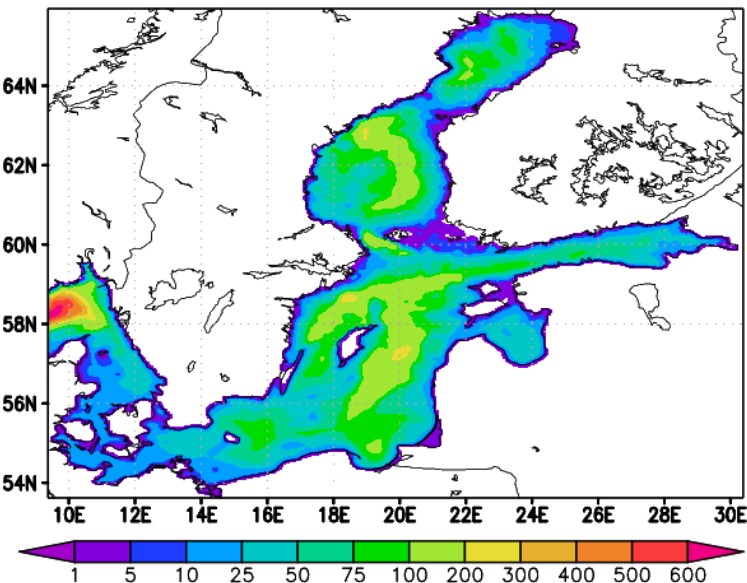

**Figure 1.** Topography of the Baltic Sea (m).

To determine the atmospheric impact (heat fluxes, fresh water and momentum on the sea surface) in the INMOM model for the Baltic Sea, the calculations used the WRF (Weather Research and Forecasting) model [53], developed by NCEP and NCAR. The period of modeling was 1 year. The Era-Interim data, with spatial resolutions $0.75° \times 0.75°$, were used as initial meteorological data for the WRF reanalysis. From the entire range of atmospheric parameters calculated using the WRF model, wind characteristics, data on air temperature, pressure, humidity, and shortwave and longwave radiation fluxes were selected. The climatic heat flow $Q^{(0)}$ was taken as a background in the cost function (8). Note that $Q^{(0)}$ has a physical meaning here. It is not only an initial guess for $Q$, but also a parameter calculated from atmospheric data and taken as the temperature boundary condition on the sea surface in the model when the model runs without the assimilation procedure.

The daily mean observations on the sea surface temperature $T_{obs}$ were obtained from the Copernicus Marine Data Store (data.marine.copernicus.eu). The daily gap-free analysis fields of SST that were used were reprocessed in the Danish Meteorological Institute, based on satellite data from infra-red radiometers [54]. The daily mean SST values were verified and interpolated to the model grid [55]. The diagonal elements of the covariance matrix $\mathcal{R}$, recalculated on the basis of the statistical properties of the observational data, were taken as weighting coefficients in the cost function when solving the data assimilation problem. Statistical characteristics (the mean and the variance) were computed separately for each day of the year on observational data for 35 years, from 1982 to 2017 [56]. Similarly, the statistical characteristics for the background error covariance matrices $\mathcal{B}$ were computed on the sea surface heat flux data for the period from 1979 to 2020, in accordance with the ERA5 reanalysis produced by the Copernicus Climate Change Service.

To illustrate the operation of the algorithm in studying sensitivity to observation errors according to (25), we considered the output function defined by

$$G(T) = \int\limits_0^{\bar{t}} dt \int\limits_\Omega W^*(x,y,t) T|_{z=0} d\Omega, \tag{26}$$

with $W^*$ being a specific weight. To estimate the mean temperature in some time interval $t_1 - \tau \leq t \leq t_1$ in a selected sea water region $\omega$, for $z = 0$, the function $W^*$ may by taken as

$$W^*(x, y, t) = \begin{cases} \dfrac{1}{\tau \mathrm{mes}\, \omega} & \text{if } t_1 - \tau \leq t \leq t_1, (x, y) \in \omega \\ 0, & \text{else}, \end{cases} \tag{27}$$

where mes $\omega$ is the area of $\omega$. Then, the output function (26) has the form:

$$G(T) = \frac{1}{\tau \mathrm{mes}\, \omega} \int\limits_{t_1-\tau}^{t_1} dt \int\limits_{\omega} T(x, y, 0, t)\, d\Omega. \tag{28}$$

The right-hand side of (28) is the mean temperature for the selected region $\omega$ in the time interval $t_1 - \tau \leq t \leq t_1$. Such output functions are of most interest in ecological problems and the theory of climate change [7].

Below, we present the results of numerical experiments with variational assimilation of the SST data for the Baltic Sea circulation model.

Figure 2 shows the daily mean SST fields for 30 May 2018, received from the Copernicus Marine Data Store, against which the model calculations were compared. The data were assimilated in the model twice daily to adjust the heat flux $Q$ in the boundary conditions. In Figures 3 and 4, the result is provided for a period of 150 days (running from 1 January to 30 May 2018). Figure 3a presents the daily mean SST field given by the model run without assimilation for the 150th day of the experiment (30 May 2018). The mean SST values, calculated using the assimilation procedure according to (5), (10)–(12), are provided in Figure 3b.

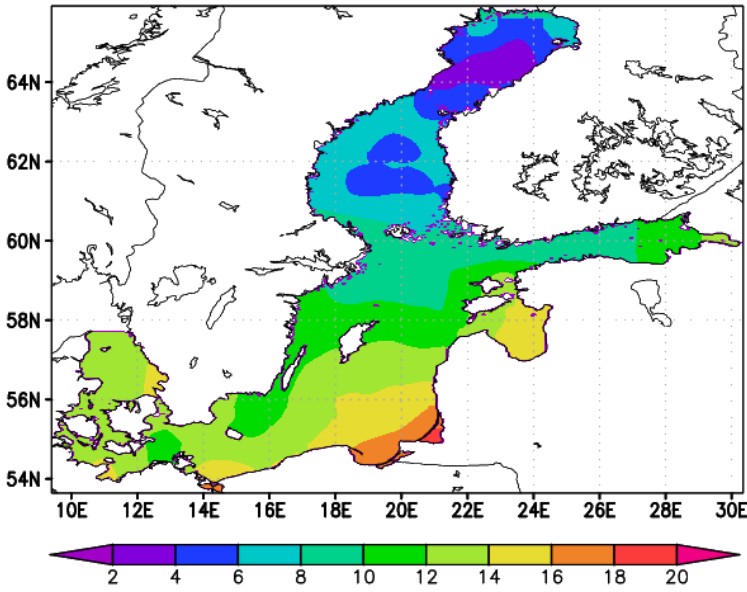

**Figure 2.** Daily mean SST observational data $T_{obs}$ (°C).

Figure 4 shows the deviation from the observational data of the daily mean sea surface temperature values calculated by the model on 30 May 2018. So, the difference between the daily mean sea surface temperature, given by the model run without assimilation ($T_{model}$), and the observational data $T_{obs}$, is shown in Figure 4a. The difference between the observational data $T_{obs}$ and the SST field obtained by assimilation ($T_{assim}$) is presented in Figure 4b, with the elements of the covariance matrices of background and observation errors used. According to Figure 4, the use of the SST variational data assimilation block

enabled results close to those actually observed from satellites, and, thereby, improved the predictive properties of the model.

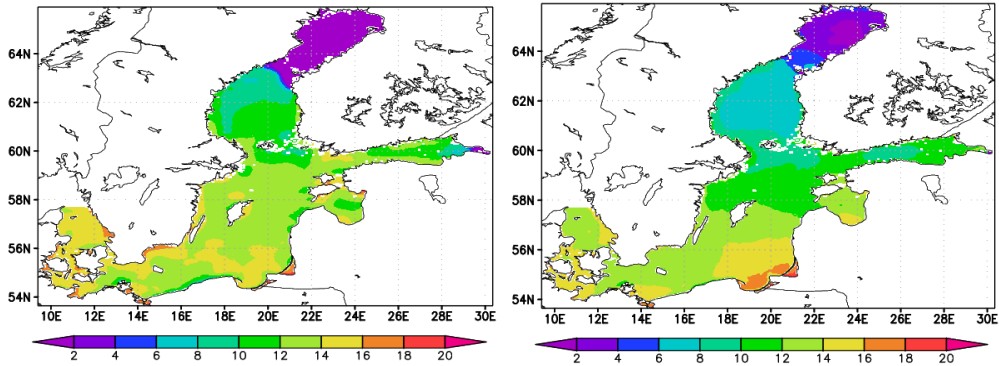

**Figure 3.** Daily mean SST (°C): (**a**) result of the model, $T_{model}$; (**b**) result with assimilation, $T_{assim}$.

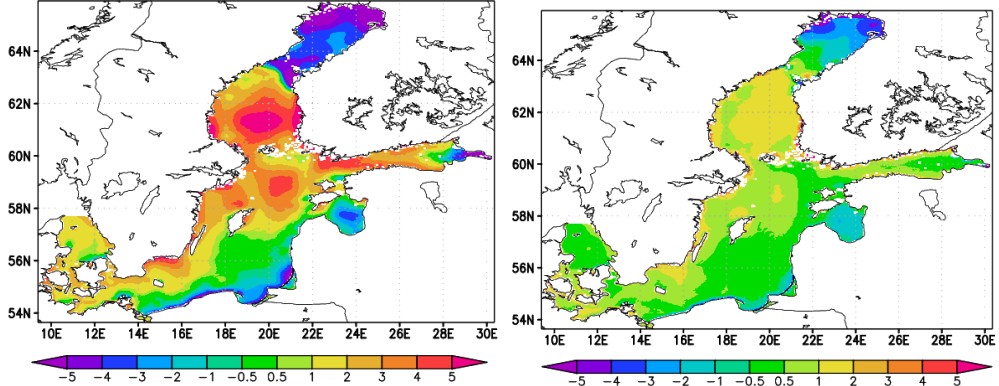

**Figure 4.** Deviation of the mean SST from observations (°C): (**a**) $T_{model} - T_{obs}$; (**b**) $T_{assim} - T_{obs}$.

Analyzing the results, it should be noted that, at the end of the experiment (30 May 2018), the model somewhat overestimated the surface temperature of the Baltic Sea in the southern part of the Gulf of Bothnia, with the deviation from the observational data reaching 5 °C, while at the southern coast of the Baltic Sea, the calculation, according to the model, had already underestimated the temperature to 4 °C, compared to the observational data. The introduction of the assimilation procedure significantly reduced these differences, and for the southern part of the Gulf of Bothnia, the deviation from observations was about 2 °C, and near the southern coast of the Baltic Sea, the deviation from observations became less than 1 °C.

Let us especially note the results of the experiment in the Gulf of Finland. In Figure 4, as a result of the run, according to the model, an underestimation of the SST was observed in the eastern part, and, on the contrary, there was an overestimation of the temperature in the western part. When calculating by means of the data assimilation procedure, it was possible to bring the temperature much closer to the observations such that, in the entire water area of the Gulf of Finland, the SST deviated from the observational data by no more than 1 °C.

We also investigated some integral SST characteristics. Table 1 shows an integral characteristic of the experiment - the value of the SST averaged over the water area of the Baltic Sea. The data were provided on the 15th day of calculation of each month. According to the table, the results of the model run with the assimilation were much closer to the values obtained from the observational data. We also noted that, at the beginning of the year, the model somewhat underestimated the sea surface temperature, while, when closer to May, the temperature had already been overestimated. Moreover, in May, the deviation from the observational data for the chosen integral characteristic reached 1.6 °C, whereas

the model run with assimilation deviated from the observational data by $-0.41$ to $0.17\,^\circ\text{C}$ throughout the experiment. This also speaks in favor of including a data assimilation block in the numerical model to improve its predictive properties.

**Table 1.** SST averaged over the entire water area on selected dates, $^\circ\text{C}$

| Date | Without Assimilation | With Assimilation | Observation Data |
|---|---|---|---|
| 15 January | 3.244 | 4.415 | 4.581 |
| 15 February | 1.779 | 2.208 | 2.038 |
| 15 March | 1.171 | 2.362 | 2.450 |
| 15 April | 3.009 | 4.564 | 4.974 |
| 15 May | 8.746 | 6.903 | 7.136 |

To estimate the sensitivity of the output function (28) to observation errors, the algorithm presented in Section 4 was used, based on (24) and (25). This algorithm revealed the regions where the gradients of the function $G(T)$ were the largest for $G(T)$ as the mean SST for a selected area, obtained by assimilation. The output function gradients calculated for 30 May 2018, according to (24) and (25), are shown in Figure 5. According to the figure, the regions having the greatest gradients were mainly near the shore of the Baltic Sea area, and in some shallow parts of the sea with depths of about 10 m (see Figure 1). Thus, $G(T)$ was most sensitive to observation errors in these areas. This result was confirmed by introducing perturbations into $T_{obs}$ and calculating, after assimilation, the output function $G(T)$ directly by (28). One explanation of this phenomenon may be the fact that in the areas with depths of about 10–20 m, rapid convection occurs in the upper mixed layer. With the assimilation of the surface temperature, information was transmitted faster to shallower depths, which, in turn, contributed to higher sensitivity to data in these places, in contrast to deeper regions.

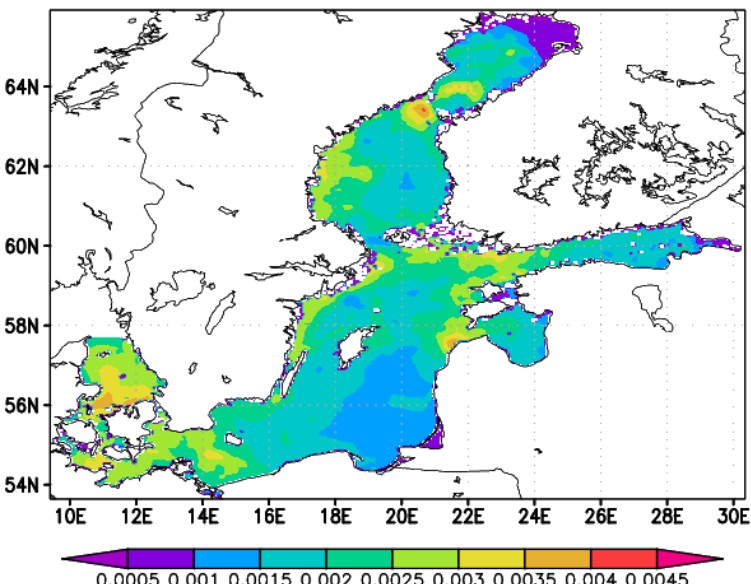

**Figure 5.** Output function gradient.

From the numerical experiments, we can see that when assimilating only the SST observational data, there was an insignificant effect on the sea surface height, currents, and water salinity. However, all hydrophysical fields remained consistent and physically reasonable after variational assimilation. Iterative algorithms for 4D-Var SST assimilation in the considered water area exhibited good convergence. In the presented experiments, the process (12) converged in less than 5 iterations.

The numerical experiments proved the efficiency of the proposed 4D-Var technology in sensitivity analysis and confirmed that the model, with its variational data assimilation procedure, improved the predictive properties. The presented algorithm (24) and (25) facilitated the estimatation of the output function sensitivity to observation errors, as a result of 4D-Var SST data assimilation.

## 6. Conclusions

The paper presents the results on 4D-Var data assimilation technology for ocean dynamics modeling with a sensitivity analysis of the model's results to observation errors. The technology is reposed on the INM RAS numerical model of ocean circulation with the splitting method and complemented by 4D-Var data assimilation with covariance matrices of background and observation errors. The technology aims to combine observational data and hydrodynamic forecasts in order to retrieve unknown model parameters. Variational data assimilation involves iterative procedures to solve inverse problems to correct sea surface heat fluxes (or other parameters) for the model under consideration. An algorithm was formulated to study the sensitivity of the model results, considered as output functions after assimilation, to the observation errors. The algorithm may be used to reveal the regions where the output function gradient is the largest, for example, for the average sea surface temperature (SST) in a selected area, obtained by assimilation. The numerical experiments demonstrated the possibility of using the reported 4D-Var data assimilation technology for modeling ocean hydrodynamic processes and evidenced fine closeness of the calculated fields to real observational data. One of the results of this study is an efficient method of heat flux correction, which could be useful for coupled ocean–atmospheric problems and could help in the development of climate and Earth system models.

**Author Contributions:** V.Z., V.A. and V.S. presented the methodology, E.P. carried out the numerical experiments, N.Z. processed the data. All authors have read and agreed to the published version of the manuscript.

**Funding:** The work was supported by the Russian Science Foundation (project 20-11-20057, studies in Sections 4 and 5), and by the Moscow Center for Fundamental and Applied Mathematics (agreement with the Ministry of Education and Science of the Russian Federation, No. 075-15-2022-286).

**Institutional Review Board Statement:** Not applicable.

**Informed Consent Statement:** Not applicable.

**Data Availability Statement:** Supporting data can be made available upon request.

**Acknowledgments:** The authors are greatly thankfull to the reviewers for valuable comments that helped improve the paper.

**Conflicts of Interest:** The authors declare no conflict of interest.

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
