# Peer review of "Four-Dimensional Variational Data Assimilation and Sensitivity of Ocean Model State Variables to Observation Errors"

_jmse, doi:10.3390/jmse11061253_

Round 1
Reviewer 1 Report
The article “4D-Var Data Assimilation and Sensitivity of Marine Characteristics to Observation Errors” by Victor Shutyaev, Vladimir Zalesny, Valeriy Agoshkov, Eugene Parmuzin, and Natalia Zakharova presents the results of the 4D-Var data assimilation technology for sea dynamics modeling with sensitivity analysis of marine characteristics to observation errors. Their results show that the use of assimilation significantly improves the modeling results.
The article is very well written and very well organized. The results are well presented, and the figures are clear and sufficient to support the conclusions. So, it should be accepted. It would have been interesting if the authors presented a brief discussion on similar works to highlight the benefits of their method; at the end, I include some articles on the subject like the one presented in this article for the discussion. I mark a couple of minimal suggestions in the PDF that might help readers of the article.
Moore, A. M., Arango, H. G., Broquet, G., Powell, B. S., Weaver, A. T., & Zavala-Garay, J. (2011). The Regional Ocean Modeling System (ROMS) 4-dimensional variational data assimilation systems: Part I–System overview and formulation. Progress in Oceanography, 91(1), 34-49.
Moore, A. M., Arango, H. G., Broquet, G., Edwards, C., Veneziani, M., Powell, B., ... & Robinson, P. (2011). The Regional Ocean Modeling System (ROMS) 4-dimensional variational data assimilation systems: Part III–Observation impact and observation sensitivity in the California Current System. Progress in Oceanography, 91(1), 74-94.
Janeković, I., Rayson, M. D., Jones, N. L., Watson, P., & Pattiaratchi, C. (2022). 4D-Var data assimilation using satellite sea surface temperature to improve the tidally-driven interior ocean dynamics estimates in the Indo-Australian Basin. Ocean Modelling, 171, 101969.
He, Z., Yang, D., Wang, Y., & Yin, B. (2022). Impact of 4D-Var data assimilation on modelling of the East China Sea dynamics. Ocean Modelling, 176, 102044.

Author Response
RESPONSE TO REVIEWER 1
The authors would like to thank the Reviewer for the time to review our paper and to provide valuable comments and suggestions. We address them in the revised version of our manuscript.
Reviewer comment:
It would have been interesting if the authors presented a brief discussion on similar works to highlight the benefits of their method; at the end, I include some articles on the subject like the one presented in this article for the discussion. I mark a couple of minimal suggestions in the PDF that might help readers of the article.
Moore, A. M., Arango, H. G., Broquet, G., Powell, B. S., Weaver, A. T., & Zavala-Garay, J. (2011). The Regional Ocean Modeling System (ROMS) 4-dimensional variational data assimilation systems: Part I–System overview and formulation. Progress in Oceanography, 91(1), 34-49.
Moore, A. M., Arango, H. G., Broquet, G., Edwards, C., Veneziani, M., Powell, B., ... & Robinson, P. (2011). The Regional Ocean Modeling System (ROMS) 4-dimensional variational data assimilation systems: Part III–Observation impact and observation sensitivity in the California Current System. Progress in Oceanography, 91(1), 74-94.
Janeković, I., Rayson, M. D., Jones, N. L., Watson, P., & Pattiaratchi, C. (2022). 4D-Var data assimilation using satellite sea surface temperature to improve the tidally-driven interior ocean dynamics estimates in the Indo-Australian Basin. Ocean Modelling, 171, 101969.
He, Z., Yang, D., Wang, Y., & Yin, B. (2022). Impact of 4D-Var data assimilation on modelling of the East China Sea dynamics. Ocean Modelling, 176, 102044.
These suggestions of the Reviewer are taken into account in the revised version of the paper.
We are greatly thankful to the Reviewer for general appreciation of our work and for useful remarks which helped us to improve the paper.
Sincerely,
Vladimir Zalesny, Valeriy Agoshkov, Victor Shutyaev, Eugene Parmuzin, Natalia Zakharova

Reviewer 2 Report
This is an interesting MS which uses a non-standard cost function and a 4Dvar approach to improve the ocean model skill. An important side product of the research is that it helps to improve the flux-correction at the sea surface- an issue which cased early coupled ocean-atmospheric models to drift from reality. This result is buried in the middle of the text but should be highlighted. My main concern is that the style and clarity of presentation requires improvement, and the text needs a lot of rewording. The MS can be published in JMSE after the following comments are taken into account.
Title and throughout the text. The term ‘Marine Characteristics’ is not common in English language oceanography. It would be better to say ‘4D-Var Data Assimilation and Sensitivity of Modelling Ocean State Variables to Observation Errors’ or ‘‘4D-Var Data Assimilation and Sensitivity of Ocean Modelling Accuracy to Observation Errors’ or similar
Lines 8-9 and throughout the text. The authors use a mathematical term ‘response function’ throughout the MS. To make the paper more appealing to the oceanographic community it is advisable to use terminology which is more commonly used in ocean modelling. The model results before Data Assimilation are called ‘forecast’ and after DA they are called ‘analysis’. The analysis is used as an amended initial condition for the next forecasting cycle.
Line 23 and Line 44. ‘..is reposed on a numerical model of marine circulation advanced…’. It is better to say ‘…is applied to a numerical model of ocean circulation developed…‘
Line 7 ‘’ sea surface heat fluxes …’ or SST? The text below discusses SST not the fluxes. Please clarify.
Line 19’.. To combine observational data..’ It is better to say ‘The combination of observational data…’
Line 25 and Line 26 ’ sea and ocean dynamics’ , ‘Ocean and sea circulation’. The word ‘sea’ is not necessary in this context.
Lines 31-42. The authors should indicate that Optimal Interpolation ( which they class as a statistical method) is equivalent to a specific variational assimilation problem as was demonstrated in (Lorenc, A. C., 1986: Analysis methods for numerical weather prediction. Quart. J. Roy. Meteor. Soc., 112, 1177-1194), see eg https://twister.caps.ou.edu/OBAN2019/3DVAR.pdf and references therein.
Line 44 and throughout the txt ‘sea dynamics’ -> ‘ocean dynamics’
Line46-47. ‘The main features of the INM RAS model, in comparison with other ocean models…’. The authors should be more precise in formulating the difference between IMS RAS and ‘other’ models. For example ‘primitive equations under hydrostastic and Boussinesq approximations in the sigma-coordinate system’ are used in a variety of current models including POM (http://www.ccpo.odu.edu/POMWEB ) and NEMO (https://www.nemo-ocean.eu).
Line 62.’ sensitivity of the model characteristics’ . You probably mean ‘sensitivity of the model results ( or outputs)’ not the characteristics of the model itself ( equations, numerical schemes etc). Please clarify.
Lines 61-63 and Lines 72-77. When discussing the sensitivity of model results to the observational errors, the authors are advised to refer and briefly compare their approach (4d-Var) with the skill of modern 3D DA methods eg [3] and (https://doi.org/10.3390/jmse11050935 )
Lines 63-64. ‘Marine characteristics are of interest in the form of response functions depending on the model variables: temperature, salinity, etc’ The meaning is unclear. Please re-word.
Line 72. ‘..was first done in ..’ -> ‘was first carried out in..’
Lines 81-82. ‘reveal the regions where the gradient of the response function is the largest, for the
averaged sea surface temperature in a selected area’ . The authors should explain why it is important to know the regions with a large gradient of the response function.
Line 83. ‘Sections of the article are organized as follows.’ It is better to say ‘The article is organized as follows.’ You are not discussing the organisation inside the sections, only how the MS is split into sections.
Line85. ‘…technique is given…’ -> ’… technique is presented…’
Line 87 ‘…surface heat fluxes..’ or SST? Please clarify. It is unlikely you have observations on sea surface fluxes, normally it is a derived variable.
Equation (1). Please specify the meaning of the following variables/parameters/operators
- (AK)2 - I can only assume that this is a kind of bi-laplacian operator)
- fS
- fT
- Theta capital,
- f3
- a_caretfi – probably the turbulent friction/viscosity coefficient
- betaT, betaS, gamma
Line 97 ‘…sea surface level function…’ ->’… sea surface level …’
Line 105. ‘QS and QT are the surface salinity and heat fluxes’ Do you mean that QT represents the sum of shortwave and longwave radiation and latent heat fluxes? Please clarify.
Lines 110-111. ‘…is the large-scale sea dynamics problem …‘ . The same equations can be used for meso- and submesoscale ocean dynamics. Please re-word.
Lines 132-133. ‘…with the aim to determine unknown model parameters [5]–[11].’ This is incorrect. DA is used to update the state variables such as temperature, salinity etc, not the model parameters such as diffusion / viscosity/ relaxation coefficients.
Line 136. ‘is an initial approximation (a background) for Q ‘. Please explain here how do you get Q0. Heat flux is not the model output but input according to your Eq (2).
Now it seems to me that one of the achievements of this study is an efficient method of ‘flux correction’ which was a significant problem in early coupled ocean-atmospheric model, see eg . https://www.researchgate.net/publication/226188804_Coupled_ocean-atmosphere_models_with_flux_correction. This is an important result which could help development of climate and Earth system models and it would be worth highlighting this fact throughout the MS (abstract, body, conclusion).
Line 176. ‘where H is the Hessian of the cost function..’. It would be beneficial for readers with ocean science background to explain the meaning of the Hessian matrix. For example to say that the Hessian matrix is a square matrix of second-order partial derivatives of a scalar field. Please also clarify what are the independent variables of the derivatives , are they latitude/longitude or something else.
Lines 202-205. When describing the model configuration for the Baltic Sea please specify the values of horizontal diffusion/viscosity coefficients, the turbulent closure scheme for calculating the vertical turbulent exchange coefficients and other configuration specific parameters.
Line 212 ‘the WRF reanalysis’ Did you use WRF in the hind cast mode or as a reanalysis ( i.e. with DA of atmospheric observations). What was the period of modelling.
Line 221-223. ‘Using the SST observational data from 1982 to 2017, the elements of the observation error covariance matrices R were computed, to be used in the minimization process.’ Please specify which method did you use to calculate the observation error covariance matrix. It is not a simple task.
Lines 223-226. Similar question for the B-matrix.
Line 291. ‘…all hydrophysical fields remain physical…’ what does it mean?
English is generally fine. Some parts of the text need to be re-written using terminology used in English language journals.
Author Response
RESPONSE TO REVIEWER 2
The authors would like to thank the Reviewer for the time to review our paper and to provide valuable comments and suggestions. We view the criticism positively, which we address in the revised version of our manuscript. Here we would like to list our responses to each item raised by the Reviewer:
Title and throughout the text. The term ‘Marine Characteristics’ is not common in English language oceanography. It would be better to say ‘4D-Var Data Assimilation and Sensitivity of Modelling Ocean State Variables to Observation Errors’ or ‘‘4D-Var Data Assimilation and Sensitivity of Ocean Modelling Accuracy to Observation Errors’ or similar
According to the suggestion of the Reviewer, we have changed the title for ‘4D-Var Data Assimilation and Sensitivity of Ocean Model State Variables to Observation Errors’.
Lines 8-9 and throughout the text. The authors use a mathematical term ‘response function’ throughout the MS. To make the paper more appealing to the oceanographic community it is advisable to use terminology which is more commonly used in ocean modelling. The model results before Data Assimilation are called ‘forecast’ and after DA they are called ‘analysis’. The analysis is used as an amended initial condition for the next forecasting cycle.
Corrected. In particular, we have replaced the term ‘response function’ with ‘output function’.
Line 23 and Line 44. ‘..is reposed on a numerical model of marine circulation advanced…’. It is better to say ‘…is applied to a numerical model of ocean circulation developed…‘
Corrected.
Line 7 ‘’ sea surface heat fluxes …’ or SST? The text below discusses SST not the fluxes. Please clarify.
Corrected.
Line 19’.. To combine observational data..’ It is better to say ‘The combination of observational data…’
Corrected.
Line 25 and Line 26 ’ sea and ocean dynamics’ , ‘Ocean and sea circulation’. The word ‘sea’ is not necessary in this context.
Corrected.
Lines 31-42. The authors should indicate that Optimal Interpolation ( which they class as a statistical method) is equivalent to a specific variational assimilation problem as was demonstrated in (Lorenc, A. C., 1986: Analysis methods for numerical weather prediction. Quart. J. Roy. Meteor. Soc., 112, 1177-1194), see eg https://twister.caps.ou.edu/OBAN2019/3DVAR.pdf and references therein.
Indicated, according to the suggestion of the Reviewer.
Line 44 and throughout the txt ‘sea dynamics’ -> ‘ocean dynamics’
Corrected.
Line46-47. ‘The main features of the INM RAS model, in comparison with other ocean models…’. The authors should be more precise in formulating the difference between IMS RAS and ‘other’ models. For example ‘primitive equations under hydrostastic and Boussinesq approximations in the sigma-coordinate system’ are used in a variety of current models including POM (http://www.ccpo.odu.edu/POMWEB ) and NEMO (https://www.nemo-ocean.eu).
Corrected.
Line 62.’ sensitivity of the model characteristics’ . You probably mean ‘sensitivity of the model results ( or outputs)’ not the characteristics of the model itself ( equations, numerical schemes etc). Please clarify.
Corrected.
Lines 61-63 and Lines 72-77. When discussing the sensitivity of model results to the observational errors, the authors are advised to refer and briefly compare their approach (4d-Var) with the skill of modern 3D DA methods eg [3] and (https://doi.org/10.3390/jmse11050935 )
Corrected, as suggested by the Reviewer.
Lines 63-64. ‘Marine characteristics are of interest in the form of response functions depending on the model variables: temperature, salinity, etc’ The meaning is unclear. Please re-word.
Corrected.
Line 72. ‘..was first done in ..’ -> ‘was first carried out in..’
Corrected.
Lines 81-82. ‘reveal the regions where the gradient of the response function is the largest, for the
averaged sea surface temperature in a selected area’ . The authors should explain why it is important to know the regions with a large gradient of the response function.
Corrected. Explanation is given in the text.
Line 83. ‘Sections of the article are organized as follows.’ It is better to say ‘The article is organized as follows.’ You are not discussing the organisation inside the sections, only how the MS is split into sections.
Corrected.
Line85. ‘…technique is given…’ -> ’… technique is presented…’
Corrected.
Line 87 ‘…surface heat fluxes..’ or SST? Please clarify. It is unlikely you have observations on sea surface fluxes, normally it is a derived variable.
Corrected. We have SST observations, but correct the heat fluxes using DA.
Equation (1). Please specify the meaning of the following variables/parameters/operators
- (AK)2 - I can only assume that this is a kind of bi-laplacian operator)
- fS
- fT
- Theta capital,
- f3
- a_caretfi – probably the turbulent friction/viscosity coefficient
- betaT, betaS, gamma
Corrected.
Line 97 ‘…sea surface level function…’ ->’… sea surface level …’
Corrected.
Line 105. ‘QS and QT are the surface salinity and heat fluxes’ Do you mean that QT represents the sum of shortwave and longwave radiation and latent heat fluxes? Please clarify.
Yes, this explanation is inserted in the text.
Lines 110-111. ‘…is the large-scale sea dynamics problem …‘ . The same equations can be used for meso- and submesoscale ocean dynamics. Please re-word.
Corrected.
Lines 132-133. ‘…with the aim to determine unknown model parameters [5]–[11].’ This is incorrect. DA is used to update the state variables such as temperature, salinity etc, not the model parameters such as diffusion / viscosity/ relaxation coefficients.
Corrected.
Line 136. ‘is an initial approximation (a background) for Q ‘. Please explain here how do you get Q0. Heat flux is not the model output but input according to your Eq (2).
Corrected. In our experiments, for Q0 the mean climatic flux is used (see also Section 5).
Now it seems to me that one of the achievements of this study is an efficient method of ‘flux correction’ which was a significant problem in early coupled ocean-atmospheric model, see eg . https://www.researchgate.net/publication/226188804_Coupled_ocean-atmosphere_models_with_flux_correction. This is an important result which could help development of climate and Earth system models and it would be worth highlighting this fact throughout the MS (abstract, body, conclusion).
Thank you very much for your fruitful comment, it is taken into account in the text of the revised version.
Line 176. ‘where H is the Hessian of the cost function..’. It would be beneficial for readers with ocean science background to explain the meaning of the Hessian matrix. For example to say that the Hessian matrix is a square matrix of second-order partial derivatives of a scalar field. Please also clarify what are the independent variables of the derivatives , are they latitude/longitude or something else.
Corrected.
Lines 202-205. When describing the model configuration for the Baltic Sea please specify the values of horizontal diffusion/viscosity coefficients, the turbulent closure scheme for calculating the vertical turbulent exchange coefficients and other configuration specific parameters.
Corrected, as suggested by the Reviewer.
Line 212 ‘the WRF reanalysis’ Did you use WRF in the hind cast mode or as a reanalysis ( i.e. with DA of atmospheric observations). What was the period of modelling.
Yes, as a reanalysis. The period of modeling was 1 year.
Line 221-223. ‘Using the SST observational data from 1982 to 2017, the elements of the observation error covariance matrices R were computed, to be used in the minimization process.’ Please specify which method did you use to calculate the observation error covariance matrix. It is not a simple task.
Lines 223-226. Similar question for the B-matrix.
Corrected in the text, with the reference.
Line 291. ‘…all hydrophysical fields remain physical…’ what does it mean?
Corrected.
We are greatly thankful to the reviewer for general appreciation of our work and for detailed and useful remarks which helped us to improve the paper.
Sincerely,
Vladimir Zalesny, Valeriy Agoshkov, Victor Shutyaev, Eugene Parmuzin, Natalia Zakharova

Reviewer 3 Report
The paper descibes an intersting and useful approach to treat sea dynamics problems with sensitivity analysis of marine characteristics to observation errors.
In the description of the results it is stated tht "the regions of the greatest gradient are discovered mainly near the shore of the Baltic Sea area, and in some shallow parts of the sea with depths of about 10 m". A deeper discussion should be added to explain this particular behaviour. How could the model be improved to reduce that gradient?
In principle well written, in some case shorter sentences could be preferred to improve understanding.
Author Response
RESPONSE TO REVIEWER 3
The authors would like to thank the Reviewer for the time to review our paper and to provide valuable comments and suggestions. We address them in the revised version of our manuscript.
Reviewer comment:
In the description of the results it is stated that "the regions of the greatest gradient are discovered mainly near the shore of the Baltic Sea area, and in some shallow parts of the sea with depths of about 10 m". A deeper discussion should be added to explain this particular behaviour. How could the model be improved to reduce that gradient?
These suggestions of the Reviewer are taken into account in the revised version of the paper.
We are greatly thankful to the Reviewer for general appreciation of our work and for useful remarks which helped us to improve the paper.
Sincerely,
Vladimir Zalesny, Valeriy Agoshkov, Victor Shutyaev, Eugene Parmuzin, Natalia Zakharova
